# Porosity Tunable Poly(Lactic Acid)-Based Composite Gel Polymer Electrolyte with High Electrolyte Uptake for Quasi-Solid-State Supercapacitors

**DOI:** 10.3390/polym14091881

**Published:** 2022-05-04

**Authors:** Chao Yang, Yuge Bai, Huan Xu, Manni Li, Zhi Cong, Hongjie Li, Weimeng Chen, Bin Zhao, Xiaogang Han

**Affiliations:** 1State Key Laboratory of Electrical Insulation and Power Equipment, School of Electrical Engineering, Xi’an Jiaotong University, Xi’an 710049, China; yc2018@stu.xjtu.edu.cn (C.Y.); baiyuge218@stu.xjtu.edu.cn (Y.B.); lmn4118004119@stu.xjtu.edu.cn (M.L.); jycz1997@stu.xjtu.edu.cn (Z.C.); lhj1124@stu.xjtu.edu.cn (H.L.); m6a8x3cwm@stu.xjtu.edu.cn (W.C.); zhaobin87@xjtu.edu.cn (B.Z.); 2Key Laboratory of Smart Grid of Shaanxi Province, Xi’an 710049, China; hihuan@cumt.edu.cn; 3School of Materials Science and Physics, China University of Mining and Technology, Xuzhou 221116, China

**Keywords:** biodegradable matrix, phase inversion, tunable porous structure, composite polymer membrane, flexible gel electrolyte, quasi-solid-state supercapacitors

## Abstract

The growing popularity of quasi-solid-state supercapacitors inevitably leads to the unrestricted consumption of commonly used petroleum-derived polymer electrolytes, causing excessive carbon emissions and resulting in global warming. Also, the porosity and liquid electrolyte uptake of existing polymer membranes are insufficient for well-performed supercapacitors under high current and long cycles. To address these issues, poly(lactic acid) (PLA), a widely applied polymers in biodegradable plastics is employed to fabricate a renewable biocomposite membrane with tunable pores with the help of non-solvent phase inversion method, and a small amount of poly (vinylidene fluoride-co-hexafluoropropylene) (PVDF-HFP) is introduced as a modifier to interconnect with PLA skeleton for stabilizing the porous structure and optimizing the aperture of the membrane. Owing to easy film-forming and tunable non-solvent ratio, the porous membrane possesses high porosity (ca. 71%), liquid electrolyte uptake (366%), and preferable flexibility endowing the GPE with satisfactory electrochemical stability in coin and flexible supercapacitors after long cycles. This work effectively relieves the environmental stress resulted from undegradable polymers and reveals the promising potential and prospects of the environmentally friendly membrane in the application of wearable devices.

## 1. Introduction

Considering the ever-expanding market demand for energy storage devices with high power, safety and long lifespan [1,2]. Researchers have focused on the development of quasi-state or solid-state supercapacitors (SCs) to avoid electrolyte leakage [3,4,5]. Compared with solid state electrolytes, gel polymer electrolytes (GPEs) applied to quasi-solid-state SCs possess better interfacial contact with electrodes and higher ionic conductivity, which are tailored to solve the safety and interfacial issues of liquid and solid SCs [6,7,8,9]. 

A well-performing GPE is expected to take up enough liquid electrolyte to ensure the electrochemical stability under high current density and long cycles [10,11]. Constructing porous structures for polymer matrix is an effective and prevalent strategy to realize high porosity and liquid electrolyte uptake (LE uptake) of GPEs for SCs [12,13]. Serving as one of the most common polymer matrixes for GPEs, poly (vinylidene fluoride-co-hexafluoropropylene) (PVDF-HFP) has attracted much research attention in the preparation of porous membrane owing to its relatively high ionic conductivity, thermal and electrochemical stabilities, and excellent mechanical properties [14,15]. While these porous membranes are still fall short of expectations to be applied in high power devices due to the comparatively low porosity and LE uptake [16]. Moreover, the production and treatment of this petroleum-based polymer will inevitably cause carbon emissions and increase environmental burden. Thus, renewable resources that serve as alternative raw materials are needed [17,18,19,20]. As an excellent commercially available representative, poly(lactic acid) (PLA), a biodegradable polyester derived from renewable biomass, such as fermented maize starch, can be used as a cost-effective substitute which will minimize the use of undegradable plastics and help achieve the goal of carbon neutrality [21,22,23]. Apart from its environmental friendliness, it is also a competitive candidate polymer electrolyte that can be applied to energy storage devices due to its easy film-forming and comparatively low crystallinity properties [24,25]. While researchers are mainly focusing on the use of aqueous-based electrolytes in PLA membranes for supercapacitors [26], which seriously limit its energy density. In view of the abovementioned issues, constructing a porous PLA-based membrane suitable for commercial organic electrolyte is necessary. Considering the excellent properties of PVDF-HFP, compositing polymers with a low amount of PVDF-HFP is a favorable tactic to improve the electrochemical performance and stability of GPEs without excessively contaminating the environment [27,28,29]. Among various preparation strategies, nonsolvent-induced phase inversion is a facile method for the rapid and safe fabrication of porous membranes, which could realize the rapid transport of ions between the electrolyte and electrodes [30,31,32,33]. By introducing few of PVDF-HFP into composite polymers and employing nonsolvent-induced phase inversion method, it is possible to develop an eco-friendly membrane with high porosity and liquid electrolyte uptake which is applicable in acetonitrile (AN) based commercial electrolyte, and the as-prepared GPE can also display excellent electrochemical stability after long cycles.

In this work, we report a novel porous PLA-based biocomposite membrane prepared by using a facile phase inversion method, which shows remarkable electrochemical properties as the GPE in the application of SCs. The blending of two polymer solutions constructs a stable porous structure, in which PLA functions as the main skeleton with excellent degradability and renewability, a few of PVDF-HFP serve as a modifier which interconnect with the loose PLA skeleton, its excellent swelling effect in AN electrolyte prevents PLA from slight solubility and ensures the structural stability of composite porous membrane, optimizing the properties of the membrane applied to the GPE. Also, the amount of PVDF-HFP used is quite small, which will not pose a threat to the environment. Easily prepared PLA-based membranes derived from an optimal amount of non-solvent exhibit high porosity (ca. 71%), a sufficient liquid electrolyte uptake (366 wt%) and an enhanced surface wettability, which possess superior rate and cyclic performance in coin type SCs. Meanwhile, the specific capacitance retention of the flexible supercapacitor (FS) is 83% at 8 mA cm^−2^, and it can be retained well (70%) under 10,000 consecutive bending cycles, suggesting that the novel biocomposite gel electrolytes is promising for sustainable and high-performance use in the future application of flexible devices.

## 2. Experimental Section

### 2.1. Materials

All the chemicals were used in pure forms as purchased. Dichloromethane (DCM) and *N,N*-dimethylformamide (DMF) were purchased from MacLean Biochemical Technology Co., LTD (Shanghai, China). Poly (vinylidene fluoride-co-hexafluoropropylene) (PVDF-HFP Mw 400,000) were purchased from Sigma-Aldrich. Poly(lactic acid) (PLA Mw 230,000) were obtained from Corbion-Purac company, Amsterdam, The Netherlands.

### 2.2. Preparation of PLA@PV Porous Biocomposite Gel Electrolyte

According to previous studies, the solution containing 15–20 wt% of PLA exhibits better membrane-forming property owing to its appropriate viscosity [34,35]. Therefore, the content of the polymer in solution is fixed at 20 wt% in this study. An amount of PLA (M_w_ = 230,000, 70−100 wt% of composite polymers) were dissolved in dichloromethane (DCM 30−70 vol.% of the total liquid) and a quantity of PVDF-HFP (30−0 wt%) were added in the other solvent *N,N*-dimethylformamide (DMF, the proportion of DMF was 30%, 40%, 50%, 60%, and 70%) to stir for 4 h, respectively. Five groups of the samples were obtained and named as 30-PLA@PV, 40-PLA@PV, 50-PLA@PV, 60-PLA@PV, and 70-PLA@PV, respectively. Then, the PVDF-HFP solution was introduced into the other solution. After stirring for 4 h at room temperature, the viscous solution was cast on the polytetrafluoroethylene (PTFE) substrate using a doctor blade with the thickness of 200 μm (Appendix A). Afterward, DCM was evaporated at room temperature for 2 h and the biocomposite membrane was immediately placed into a vacuum oven at 60 °C for 10 h to evaporate the residual solvent. For comparative studies, PLA and PVDF-HFP membranes were prepared using similar methods. GPEs were used to assemble supercapacitors after the porous membranes were immersed into 1M MeEt_3_NBF_4_/acetonitrile (AN) organic electrolyte.

### 2.3. Characterization

The microstructure of the membranes was observed by scanning electron microscopy (SEM, Phenom Pro X, Thermo Fisher Scientific, Waltham, MA, USA), and the Energy Dispersive X-ray (EDX) analysis was carried out by this machine. The wettability test was conducted with a contact angle goniometer (DSA100, Krüss GmbH, Hamburg, Germany). Fourier transform infrared (FTIR) spectra of (PVDF-HFP, PLA, and 60-PLA@PV) membranes were recorded using a Nicolet IN10+IZ10 FTIR spectrophotometer (Thermo Fisher Scientific, Waltham, MA, USA) in transmission/reflection mode. The thermal stability of the membranes was tested by thermal gravimetric analysis (TGA) at a heating rate of 10 °C min^−1^ from 25 to 600 °C under N_2_ atmosphere and the mass percentage of each component was recorded under air atmosphere. The dimensional stability was measured by the muffle furnace (SL-1100, Haoyue Furnace, Shanghai, China) under 150 °C for 10 min at ambient condition. The transition temperature of membranes was performed by Differential scanning calorimetry instrument (DSC) (DISCOVER DSC250, TA Instruments, New Castle, DE, USA), and samples were heated from 0 to 200 °C under N_2_ atmosphere at the rate of 10 °C·min^−1^. The liquid electrolyte uptake ratio was evaluated according to the Equation (1):(1)$$\mathrm{Electrolyte}\;\mathrm{Uptake}\;\left(\%\right)=\frac{W-W_{0}}{W_{0}}\times 100\%$$
where *W_0_* and *W* are the weight of polymer membrane before and after soaking in liquid electrolyte, respectively. The porosity of biocomposite membranes was measured by the n-butanol absorbing method and computed by Equation (2)
(2)$$\mathrm{Porosity}\;\left(\%\right)=\frac{W_{f}-W_{i}}{\rho \;V_{0}}\times 100\%$$
where $W_{i}$ and $W_{f}$ are weights of dry and butanol absorbed membrane samples, respectively, the density of butanol is termed as ρ.

The liquid we used for testing contact angle measurements was 1M MeEt_3_NBF_4_/acetonitrile (AN) organic electrolyte.

### 2.4. Electrochemical Measurement

Electrodes of cell were prepared by mixing YP-50F commercial active carbon (AC) powder (80%) with 10 wt% Super P carbon black and 10 wt% PVDF into amount of *N*-methyl-2-pyrrolidinone (NMP). The coin cells (CR 2025) were assembled in an argon-filled glovebox with GPE (the diameter of 14 mm) for further testing.

Prior to assemble flexible devices, the square electrodes were prepared according to our previous work using AC [36]. The flexible supercapacitors consisted of two symmetric AC electrodes (20 mm × 20 mm) sandwiching a piece of GPE (25 mm × 25 mm). Electrolyte for 3 V and 3.5 V test is prepared by adding 1.5 M [PY_14_][BF_4_] into 1 M MeEt_3_NBF_4_/acetonitrile (AN) organic electrolyte. To ensure close contact of the interfaces between the electrodes and electrolyte, the assembled flexible SCs were pressed and sealed by thermoplastic films.

The electrochemical performance of the as-fabricated SCs were determined in a two-electrode system using electrochemical impedance spectroscopy (EIS), cyclic voltammetry (CVand linear sweep voltammetry (LSV) experiments performed with SS/GPE/Ag cell (SS: Stainless Steel) (10 mV s^−1^) on an electrochemical workstation (BioLogic VMP3, Seyssinet-Pariset, France). The EIS measurements were tested by applying an AC voltage with 10 mV amplitude at frequencies ranging from 100 kHz to 100 mHz for coin cell, 1 MHz to 100 mHz for flexible device. Galvanostatic charge/discharge (GCD) cycling, rate and cycling performance were performed by an Arbin MSTAT4 electrochemical station (College Station, TX, USA) at room temperature.

## 3. Results and Discussion

### 3.1. Synthesis of Porous Gel Polymer Electrolyte

Figure 1a provides the schematic illustration of preparing the composite of PLA and PVDF-HFP (PLA@PV) porous biocomposite gel. PLA is regarded as the main component of the membrane to build polymer matrix attributed to its excellent degradation behavior and great film-forming property [37,38]. PVDF-HFP functions as a modifier in the biocomposite membrane [29]. A common and facile method of nonsolvent-induced phase inversion is employed to produce pores which can facilitate the ion transportation of the GPE. In particular, *N,N*-dimethylformamide (DMF) is selected as the non-solvent, because it solves the insolubility issue of PVDF-HFP in dichloromethane (DCM) and induces phase inversion during the preparation of the composite membrane by adjusting the ratio between the solvent and nonsolvent in the formation of desired porous samples [34,35]. Two steps are involved in this process. First, PLA constitutes the main skeleton accompanied by the rapid volatilization of DCM. Then, the PVDF-HFP solution is evenly dispersed to embellish the interior of the PLA skeleton for stabilizing the structure. The porous membrane arises from the slow volatilization of DMF droplets under high temperature. All in all, the synergistic effect of two solutes (PLA and PVDF-HFP) and two solvents (DCM and DMF) forms a functional membrane. Figure 1b_1_–b_4_ provides the digital graphs of different porous membranes with various ratios of PLA and PVDF-HFP. As shown in Figure 1b_1_, the porous pure PLA membrane becomes brittle after being immersed into AN-based electrolyte, and its surface becomes rough, similar to the dense PLA membrane in Appendix A). This result is attributed to the slight solubility of PLA in AN. By introducing PVDF-HFP into the membrane and adjusting the ratio of PVDF-HFP and PLA from 1:9 to 3:7, the membrane gradually becomes stable and maintains its integrity (Figure 1b_2_–b_4_). Based on the carbon neutralization goal, the GPE is stable enough for application when the ratio of the two polymers is 2:8.

### 3.2. Morphology and Structure Characterization, Porosity, Liquid Uptake, and Affinity

Based on the ratio of PVDF-HFP and PLA is fixed at 2:8, the as-prepared GPEs determined by different proportions of nonsolvent are displayed by digital images. According to Appendix A), excessive or deficient nonsolvent can lead to the precipitation of polymers in the slurry. Samples with proportions of 30%, 40%, 50%, 60%, and 70% DMF were marked as 30-PLA@PV, 40-PLA@PV, 50-PLA@PV, 60-PLA@PV, and 70-PLA@PV, respectively. As shown in Figure 2a, compared to other intact samples with grainy surfaces, the surface of 60-PLA@PV membrane is the smoothest, which resulted from the solubility difference of PLA in the mixed solvent. To further observe the effect of ratio between nonsolvent and solvent on the porous membrane morphology, micrographs of membranes are characterized by SEM in Figure 2b_1_–e. The membrane with boundaries and bulk particles observed on the surface gradually transform into a mesh structure with increasing proportion of DMF from 30% to 70%. The cross-sectional morphology of samples gradually changes from dense to poriferous structures with increasing thickness as shown in Figure 2c_1_–c_5_. Notably, the 60-PLA@PV sample shows a relatively flat surface with macropores of about 0.5−1.5 μm (Figure 2d), and its cross section shows a multi-level porous morphology that contribute to the improvement of ionic transport. According to the EDX mapping images of 30-PLA@PV (Appendix A), PVDF-HFP tends to appear at the boundary of bulk PLA, which is formed during the blending of two mutually insoluble polymer solutions. And PVDF-HFP functions as a modifier that interconnects with PLA to strengthen the skeleton. Increasing the amount of DMF promotes the wide and uniform distribution of PVDF-HFP solution in the PLA skeleton, thereby enabling PVDF-HFP to evenly modify the porous structure with a flat surface and a suitable aperture. By contrast, the porous PLA membrane without PVDF-HFP exhibits a discontinuous flocculent structure (Appendix A), this further validates the modifying effect of PVDF-HFP on the construction of porous composite membranes. The 70-PLA@PV sample with uneven macro pores on the surface and loose structure caused by excessive nonsolvent (Figure 2e) may result in weak mechanical strength and serious LE leakage under deformation. Porosity is a pivotal parameter in evaluating the porous structure of membranes. Figure 2f demonstrates that increasing the proportion of DMF in the slurry can contribute to the higher porosity of the membranes, and all of the prepared porous membranes possess much higher porosity than the pure PVDF-HFP membrane (ca. 11%).

The porosity of the membrane is further assessed by the LE uptake when the membrane is saturated in the electrolyte as shown in Figure 3a. The LE uptake of 50-PLA@PV, 60-PLA@PV and 70-PLA@PV samples can reach to 255, 366 and 420 wt%, respectively. For comparison, the LE uptake of the PVDF-HFP counterpart is only 132%, which indirectly explains its inferiority in cycling tests. By combining Figure 2f and Figure 3a, it becomes evident that porosity is positively correlated with LE uptake (Figure 3b). Considering the key role of porous structure in improving the wetting speed and compatibility between membranes and LE in the performance of SCs, the LE contact angles on membranes displayed in Figure 3c shows that samples with higher porosity presents better wetting ability. Moreover, the contact angles of 60-PLA@PV and 70-PLA@PV samples drop to lower than 15° after 10 s, which are much smaller than that of PVDF-HFP (28.8°), indicating the faster solvent infiltration ability of the as-prepared porous membranes compared with the dense counterparts.

### 3.3. Physical and Chemical Properties

The component of the as-prepared biocomposite membrane was confirmed by comparing the FT-IR spectra of PVDF-HFP, PLA, and 60-PLA@PV samples (Appendix A). The retention of the FT-IR characteristic peaks of PLA and PVDF-HFP manifests the absence of any form of chemical reaction between the two polymers. Besides, TGA results (Appendix A) can also prove the composition of the porous membrane. As the thermal decomposition is tested from 25 °C to 600 °C, the pure PLA and pure PVDF-HFP start to degrade at 317.5 °C and 425.0 °C, respectively, the porous membrane has the above two steps under N_2_ flow. Subsequently, The PVDF-HFP content in 60-PLA@PV is 20 wt% verified by TGA under air atmosphere. It is worth stressing that thermal stability is significant for the evaluation of polymer electrolytes in SCs. In the application of electrochemical device, the accidental over-charge/discharge, internal or external heavy circuit, will bring serious safety hazards caused by thermal failure of over-heated commercial separators [39,40]. So the comparatively high decomposition temperature (over 300 °C) of biocomposite membrane guarantees the high safety of devices. Digital images and SEM morphologies of three membranes after being heated at 150 °C for 1 h are compared as shown in Figure 4a. The surface of PVDF-HFP and PLA membranes become uneven after heating, and PLA membrane turns to corrugated and brittle, which brings a huge risk in thermal runaway of flexible supercapacitors. On the contrary, the morphology of 60-PLA@PV sample is well maintained, which can effectively prevent the occurrence of thermal runaway. The enhanced stability of 60-PLA@PV is likely originated from the special porous structure which provides enough space to avoid the thermal deformation [41]. As the ions of LEs migrate through the disordered amorphous phase of polymers [42,43], the crystalline entities should have influence on the ionic conductivity. Upon incorporation of PVDF-HFP [14], the melting peaks of 60-PLA@PV slightly shift to the left from pure PLA melting temperature to 150 °C (Figure 4b), indicating that the molten state of 60-PLA@PV membrane contributes to the maintenance of its morphology under 150 °C, and the crack of pure PLA membrane is owing to its higher melting temperature and lower fracture toughness [44]. Also, the membrane with lower melting temperature has more active segment movement which can facilitate the ionic transport. Furthermore, the ionic conductivity of different samples is compared in Figure 4c, composite samples display much higher values compared to PVDF-HFP counterpart, and, as the porosity of the sample rises, the ionic conductivity gets higher.

### 3.4. Electrochemical Performances of Coin Type SCs

Coin type SCs were assembled with commercial active carbon and the as-prepared GPEs to evaluate the electrochemical performance of SCs (Figure 5 and Appendix A). EIS was performed to investigate the intrinsic internal resistance and charge transfer properties of the devices [45,46]. Porous samples possess low intrinsic internal resistance and charge transfer resistance, which indicates that the porous GPEs have tight interfacial contact with electrodes and rapid charge transfer and ionic transport, as displayed in Appendix A). Moreover, the rectangular CV curves of 60-PLA@PV sample with varied scan rates and its wider electrochemical window (~4.5 V) compared to PVDF-HFP sample (~3.7 V) represents the characteristics of an ideal EDLC (Appendix A) [47]. The capacitive performance is investigated with GCD cycling experiments at various current densities of 1, 10, and 20 A g^−1^. The symmetric triangle feature of GCD curves (Figure 5a) indicates excellent electrochemical reversibility and capacitive behavior. The 60-PLA@PV exhibits smaller IR drop (0.29 V) and much higher specific capacitance (83 F g^−1^) compared with the other two samples (0.50 V for 50-PLA@PV and 0.73 V for 70-PLA@PV) under the current density of 20 A g^−1^. The gravimetric capacitance difference of three GPE-based SCs (Figure 5b) increases rapidly with gradual increase in the current density, indicating the lower charge transfer resistance of 60-PLA@PV, as shown in Appendix A). Usually, the porous samples with high LE uptake can provide more ions to ensure long cycles, thereby demonstrating excellent cycling stability. While the 70-PLA@PV sample with lower capacitance retention may be attributed to the uneven macropores on the surfaces, which increases the IR drop. On the contrary, the 60-PLA@PV sample with much smaller pores on the surface shows excellent performance by maintaining over 90% capacitance at 2.5 V with the current density of 1 A g^−1^ (Figure 5c) after continuous 10,000 cycles. The Ragone plot shown in Appendix A) compares the energy density and power density of different samples, it demonstrates that 60-PLA@PV sample possesses high power with higher energy density maintained, while the power density and energy density cannot be balanced in other samples. Figure 5d exhibits the integrity of the 60-PLA@PV GPE at both macro and micro levels after charging/discharging for 10,000 cycles. The nearly unchanged morphology of the GPE proves its electrochemical stability.

### 3.5. Electrochemical Performances of Flexible Supercapacitors

Considering the good deformability of 60-PLA@PV membrane, the well-behaved 60-PLA@PV GPE was assembled in the flexible SC (Figure 6a) and designated as FS60-PLA@PV to assess its electrochemical performance. For comparison, a supercapacitor with the same electrodes and PVDF-HFP as the electrolyte was also assembled (i.e., FS-PVDF-HFP). Appendix A) shows the CV curves of FS60-PLA@PV at different scan rates (5−100 mV s^−1^) with a potential window of 0−3 V owing to its superior electrochemical stability. The FS60-PLA@PV shows an almost rectangular shape, even at a high scan rate (500 mV s^−1^) under bending state with radius of 12.5 mm (Figure 6b). In addition, the FS60-PLA@PV displays a standard triangular shape according to the GCD profiles in Figure 6c, thereby revealing the ideal EDLC behavior [48]. Capacitance retention under high current is a primary indicator that verifies the rate performance of a porous electrolyte. As shown in Figure 6d, the capacitance is 77.6 mF cm^−2^ at 0.5 mA cm^−2^, and 64.6 mF cm^−2^ remains even at a current density of 8 mA cm^−2^. Moreover, the lower charge transfer resistance (R_ct_) of FS60-PLA@PV (0.2 Ω) (Appendix A) compared with FS-PVDF-HFP (0.5 Ω) under deformation shows that the as-prepared GPE allows rapid ion transportation during the charge–discharge process [49]. The cycling stability was evaluated under bending status between 0 and 3 V at a current density of 0.5 mA cm^−2^. The preferable capacitance retention of the FS60-PLA@PV (70% after 10,000 cycles) compared to FS-PVDF-HFP (capacitance retention of 10%) in Figure 6e exhibiting the excellent electrochemical reversibility of the as-prepared sample. To demonstrate the applicability of the full configuration, the photographs of an illuminated LED are presented when the flexible SC is bended before cycling and unfolded after long cycles, as shown in the inset of Figure 6e. Furthermore, regarding the relatively high voltage window of the biocomposite electrolyte, the operating potential of flexible devices is extended to 3.5 V. The resulting CV curves demonstrate the high-voltage and high flexibility of the resultant SC with this GPE (Appendix A). Overall, the biocomposite gel electrolyte, which was placed in cell and flexible SCs, exhibited superior electrochemical and flexible performance even under high voltage potential, thereby offering a promising prospect for replacing the traditional non-degradable plastic electrolyte.

## 4. Conclusions

A porous PLA-based biocomposite gel polymer electrolyte in the application of supercapacitors (SCs) is successfully prepared by using a phase inversion strategy, thereby effectively ameliorating the ungreen property of pure PVDF-HFP membrane and strengthening the stability of PLA membrane employed in organic electrolyte. By introducing few of PVDF-HFP into the PLA skeleton to modify the membrane with interconnected porous structure and preferable aperture, the GPE overcomes the slight solubility in acetonitrile (AN) based solution and entitled to stable structure, favorable porosity, and high liquid electrolyte uptake. Owing to this unique porous structure, the GPE of 60-PLA@PV displays superb rate stability with 100% capacitance retention at the current density of 20 A g^−1^ and 90% capacitance retention at 1 A g^−1^ after 10,000 cycles in coin type SCs. Moreover, comparing to PVDF-HFP counterpart, the outstanding flexibility of 60-PLA@PV GPE enables the flexible device with AC electrodes to exhibit superior capacitance retention (a superior rate of 83% capacitance retention even at 8 mA cm^−2^ under 3 V), remarkable cycling stability (70% capacitance retention after 10,000 cycles) and high voltage window under deformation. The well-behaved GPE prepared in this work provides new insights into the development of biodegradable polymer electrolyte for practical applications to wearable devices and push the replacement of petrochemical-based polymer electrolytes with renewable polymers as a decisive step forward.

## Figures and Tables

**Figure 1 polymers-14-01881-f001:**
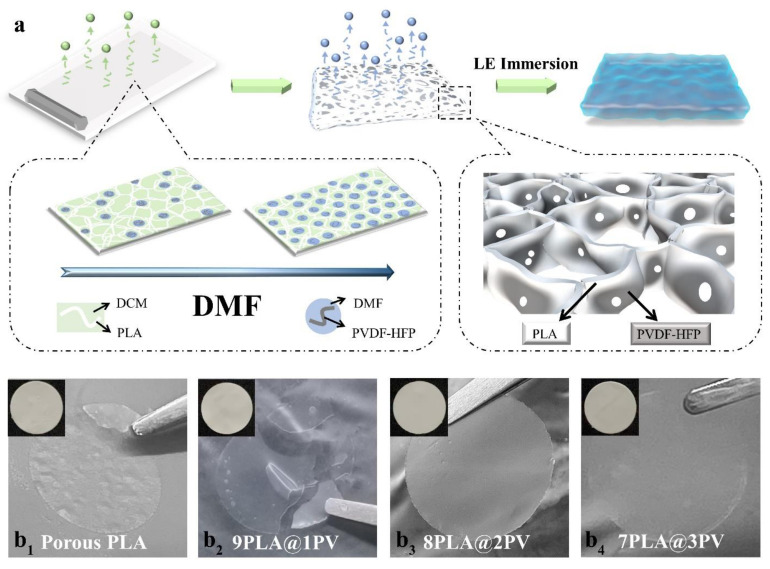
(**a**) Schematic illustration for the preparation of GPE, (**b_1_**–**b_4_**) the digital graphs of the GPEs with different PVDF-HFP proportions.

**Figure 2 polymers-14-01881-f002:**
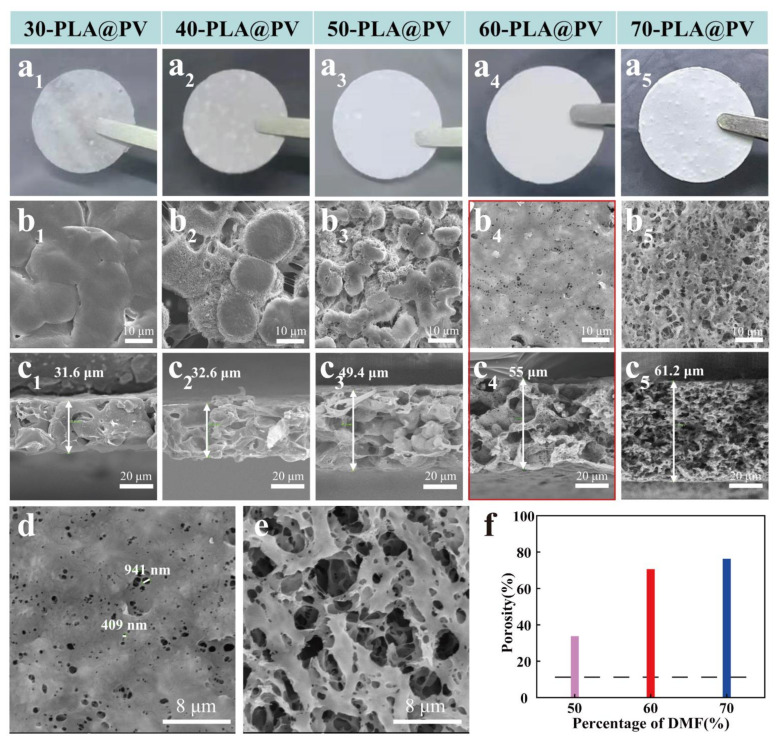
(**a1**–**a5**) Digital images of five membranes prepared with various ratios of DMF and DCM. (**b1**–**e**) SEM images: surface (**b_1_–b_5_**,**d**,**e**) and cross-section (**c_1_–c_5_**) images of five membranes. (**f**) Porosity of four membranes.

**Figure 3 polymers-14-01881-f003:**
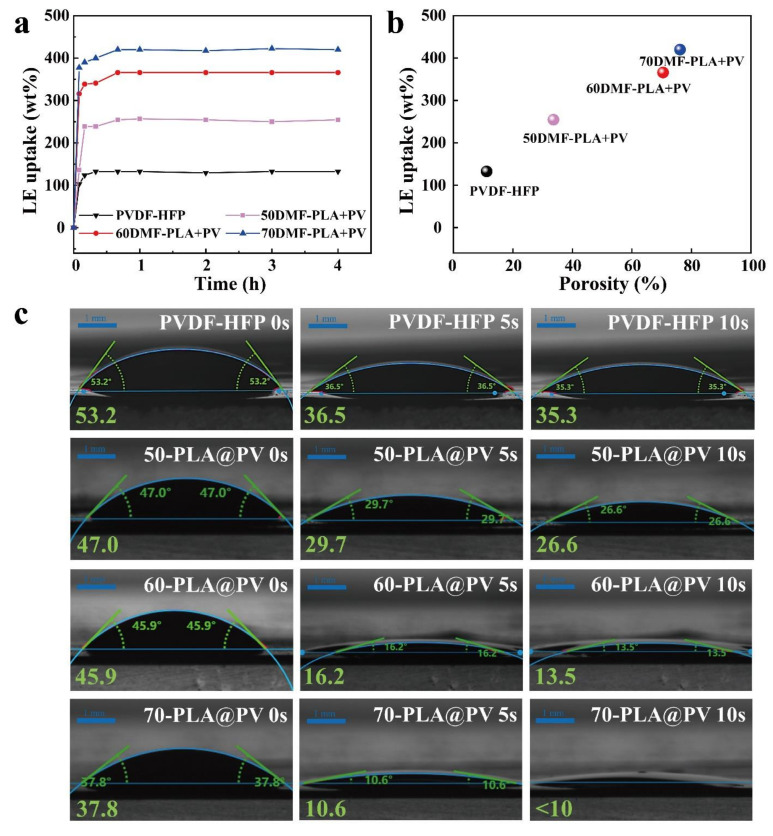
Properties of the membranes: (**a**) Liquid electrolyte uptake of five polymer membranes. (**b**) Liquid electrolyte uptake of membranes with different porosities. (**c**) Images of LE contact angles on PVDF-HFP, and 50-PLA@PV, 60-PLA@PV, and 70-PLA@PV at different times.

**Figure 4 polymers-14-01881-f004:**
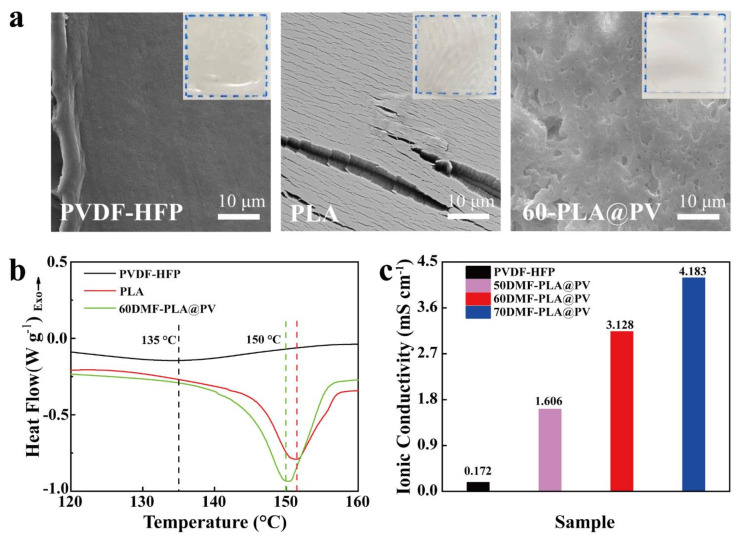
Characterization of physical and chemical properties of composite membranes: (**a**) Digital images and SEM morphologies of the three membranes after thermal treatment. (**b**) DSC thermograms of PLA, PVDF-HFP and the porous 60-PLA@PV. (**c**) Ionic conductivities of different GPEs.

**Figure 5 polymers-14-01881-f005:**
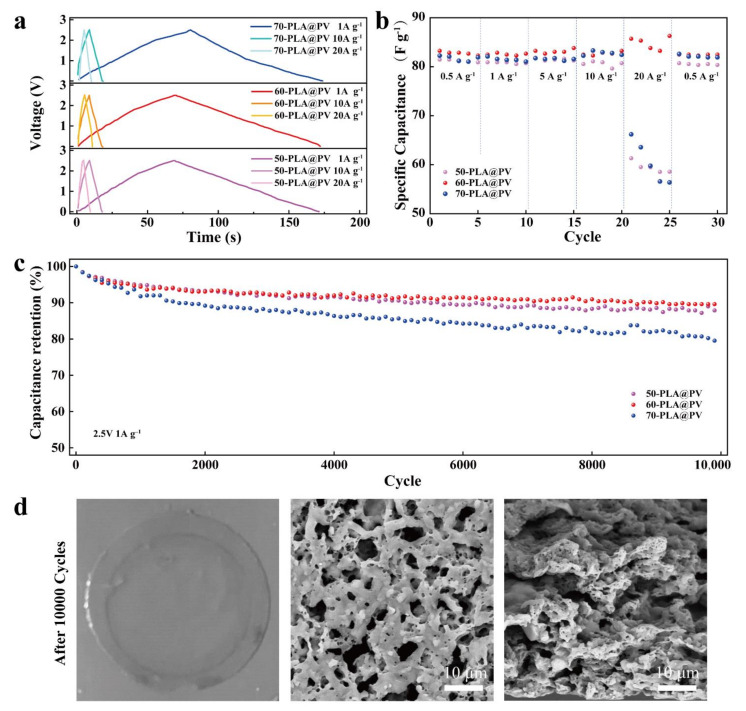
Electrochemical performance of coin type SCs under 2.5 V: (**a**) Galvanostatic charge/discharge profile. (**b**) Specific capacitance plots at various current densities. (**c**) Cycling performance at 1 A g^−1^ for 10,000 cycles. (**d**) Digital and SEM graphs (surface and cross section) of the GPE after 10,000 cycles.

**Figure 6 polymers-14-01881-f006:**
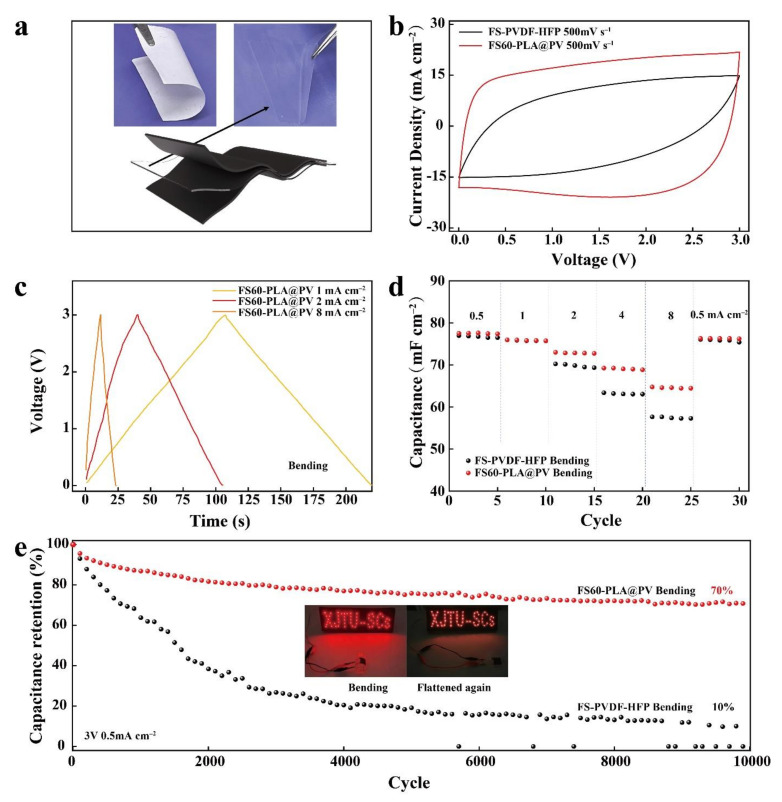
Flexible supercapacitor: (**a**) Digital images of flexible membrane before and after LE immersion. (**b**) Comparison of CV curves at high scan rates of FS. (**c**) Galvanostatic charge/discharge profiles. (**d**) Specific capacitance with increasing current densities (3 V). (**e**) Cycling performance at 0.5 mA cm^−2^ (3 V).

## Data Availability

All the data required are reported in this manuscript and Appendix A.

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
