# Peer review of "Porosity Tunable Poly(Lactic Acid)-Based Composite Gel Polymer Electrolyte with High Electrolyte Uptake for Quasi-Solid-State Supercapacitors"

_polymers, 2022, doi:10.3390/polym14091881_

Round 1

Reviewer 1 Report

I enjoyed reading this manuscript and found the motivation, characterization and materials somewhat compelling. There are a few points that should be considered prior to publication, listed below.

  1. The experimental section lacks some details including the details as to the sourcing of all of the species and polymers. For example more details on the PLA and PVDF used.
  2. I didn't understand why n-butanol was used for the porosity determination in lieu of the electrolyte itself.
  3. I'm assuming, but it's not described, that the contact angle measurements were with DI water?
  4. The naming structure felt a little inconsistent between Figure 1 and 2. In figure 1 the pre-scripts on PLA and PV are the relative composition. Whereas in Figure 2, the pre-script to the name is the solvent content. The section doesn't describe what the PLV-PV content used was. I'm assuming it was 2:8 identified in section 3.1; but it was unclear.
  5. On page 8 the shift in the Tm on Figure 4b is described as being related to "decrease in crystallinity" which isn't necessarily true. A change in the total enthalpy of melting observed would correspond to a change in crystallinity (something the authors should consider extracting and comparing), where the melting temperature shift itself has more to do with the crystallization itself.

Author Response

Thank you very much for allowing us to revise our manuscript for publication in Polymers. We would like to show you our point-by-point responses to revision. In the marked copy of the revised manuscript, the revisions are highlighted in red for Reviewer 1

Reviewer 1

I enjoyed reading this manuscript and found the motivation, characterization and materials somewhat compelling. There are a few points that should be considered prior to publication, listed below.

Author Reply: Thanks for your positive evaluation about our work.

  1. The experimental section lacks some details including the details as to the sourcing of all of the species and polymers. For example more details on the PLA and PVDF used.

Author Reply 1: Thank you for pointing out the missing information of the materials. We have completed the detailed information of the materials which are highlighted in red color.

  1. I didn't understand why n-butanol was used for the porosity determination in lieu of the electrolyte itself.

Author Reply 2: Thank you. You raised a good question regarding the function of n-butanol in the porosity test. The surface tensions of n-butanol and the organic electrolyte are basically equal, and the toxicity of n-butanol is much lower than the organic electrolyte, that’s the reason of using n-butanol for porosity test. Besides, several published research papers have also applied n-butanol for porosity test. For example: (1). Yang, C.; Sun, M.; Wang, X.; Wang, G., A novel flexible supercapacitor based on cross-linked pvdf-hfp porous organogel electrolyte and carbon nanotube paper@π-conjugated polymer film electrodes. ACS Sustainable Chemistry & Engineering 2015, 3 (9), 2067-2076. (2). Li, D.; Shi, D.; Xia, Y.; Qiao, L.; Li, X.; Zhang, H., Superior thermally stable and nonflammable porous polybenzimidazole membrane with high wettability for high-power lithium-ion batteries. ACS Applied Materials & Interfaces 2017, 9 (10), 8742-8750.

  1. I'm assuming, but it's not described, that the contact angle measurements were with DI water?

Author Reply 3: Thank you for pointing out this question. We are sorry for the missing information of the liquid used for the contact angle measurements. Actually the liquid we used for testing was 1M MeEt3NBF4/ acetonitrile (AN) organic electrolyte, and we have added this information in the experimental section with red highlight.

  1. The naming structure felt a little inconsistent between Figure 1 and 2. In figure 1 the pre-scripts on PLA and PV are the relative composition. Whereas in Figure 2, the pre-script to the name is the solvent content. The section doesn't describe what the PLV-PV content used was. I'm assuming it was 2:8 identified in section 3.1; but it was unclear.

Author Reply 4: We really appreciate your suggestion, and the ratio of PVDF-HFP and PLA in section 3.2 is indeed missing, which has been added now to emphasize the 2:8 ratio of two polymers with red highlights.

  1. On page 8 the shift in the Tm on Figure 4b is described as being related to "decrease in crystallinity" which isn't necessarily true. A change in the total enthalpy of melting observed would correspond to a change in crystallinity (something the authors should consider extracting and comparing), where the melting temperature shift itself has more to do with the crystallization itself.

Author Reply 5: Thank you for pointing out this mistake, we have revised our description of DSC curves of two membranes in red highlight.

Reviewer 2 Report

This study reported a novel porous PLA-based biocomposite membrane prepared by using a facile phase inversion method, which shows remarkable electrochemical properties as the GPE in the application of SCs.

The authors made an in-depth description of the literature, which is an introduction to the manuscript. The scientific literature is correctly cited. The research experiment was properly planned and executed. The test results were correctly interpreted and statistically evaluated. The conclusions from the conducted research were correctly formulated.

Some remarks:

- which are the atmosphere for DSC analysis and the flow rate.

Important issue:

The supplementary file is not uploaded in the system. All S Figures are not in the uploaded manuscript.

I will revised my decision after I will see the all submission files.

Author Response

Thank you very much for allowing us to revise our manuscript for publication in Polymers. We would like to show you our point-by-point responses to revision.  

Reviewer 2

This study reported a novel porous PLA-based biocomposite membrane prepared by using a facile phase inversion method, which shows remarkable electrochemical properties as the GPE in the application of SCs.

The authors made an in-depth description of the literature, which is an introduction to the manuscript. The scientific literature is correctly cited. The research experiment was properly planned and executed. The test results were correctly interpreted and statistically evaluated. The conclusions from the conducted research were correctly formulated.

I will revised my decision after I will see the all submission files.

Author Reply: Thanks for your affirmation of our work.

  1. which are the atmosphere for DSC analysis and the flow rate.

Author Reply 1: Thank you for reminding us to supply the information of the atmosphere used for DSC analysis. We’ve added the Nitrogen atmosphere and flow rate in the DSC experimental section.

  1. The supplementary file is not uploaded in the system. All S Figures are not in the uploaded manuscript.

Author Reply 2: Thank you for pointing out this important issue. The missing supporting file may attribute from the uploaded compressed package. We’ve uploaded the supporting file again.
